# Patients Treated for HCV Infection and Listed for Liver Transplantation in a French Multicenter Study: What Happens at Five Years?

**DOI:** 10.3390/v15010137

**Published:** 2022-12-31

**Authors:** Lucy Meunier, Mohamed Belkacemi, George Philippe Pageaux, Sylvie Radenne, Anaïs Vallet-Pichard, Pauline Houssel-Debry, Christophe Duvoux, Danielle Botta-Fridlund, Victor de Ledinghen, Filomena Conti, Rodolphe Anty, Vincent Di Martino, Marilyne Debette-Gratien, Vincent Leroy, Theophile Gerster, Pascal Lebray, Laurent Alric, Armand Abergel, Jérôme Dumortier, Camille Besch, Helene Montialoux, Didier Samuel, Jean-Charles Duclos-Vallée, Audrey Coilly

**Affiliations:** 1Montpellier Saint Eloi University Hospital, 80 Avenue Augustin Fliche, 34090 Montpellier, France; 2Nouvelles Technologies, AESIO Santé, 34070 Montpellier, France; 3Croix-Rousse Hospital, Lyon University Hospital, 103 Grande Rue de la Croix-Rousse, 69004 Lyon, France; 4Cochin Hospital, Public Hospitals of Paris, 27 Rue du Faubourg Saint-Jacques, 75014 Paris, France; 5Pontchaillou University Hospital, 2 Rue Henri le Guilloux, 35000 Rennes, France; 6Henri-Mondor University Hospital, Public Hospitals of Paris, 51 Avenue du Maréchal de Lattre de Tassigny, 94010 Créteil, France; 7Marseille Public Hospital, Timone University Hospital, 264 Rue Saint Pierre, 13005 Marseille, France; 8Hepatology and Liver Transplantation Unit, Haut-Lévêque Hospital, Bordeaux University Hospital, 33600 Pessac, France; 9Pitié-Salpêtrière University Hospital, Public Hospitals of Paris, 47-83 Boulevard de l’Hôpital, 75013 Paris, France; 10Archet 2 Hospital, Nice University Hospital, 151 Route de Saint-Antoine, 06200 Nice, France; 11Besançon Regional University Hospital, 3 Boulevard Alexandre Fleming, 25000 Besançon, France; 12Limoges University Hospital, 2 Avenue Martin Luther King, 87000 Limoges, France; 13Service d’Hépato-Gastroentérologie, Pôle Digidune, CHU Grenoble Alpes, 38700 La Tronche, France; 14Rangueil Hospital, Toulouse 3 University Hospital, 31000 Toulouse, France; 15Gabriel-Montpied Hospital, Clermont-Ferrand University Hospital, 58 Rue Montalembert, 63000 Clermont-Ferrand, France; 16Edouard Herriot Hospital, Lyon University Hospital, 5 Place d’Arsonval, 69003 Lyon, France; 17Hautepierre Hospital, Strasbourg University Hospital, 1 Avenue Molière, 67200 Strasbourg, France; 18Rouen University Hospital, 37 Boulevard Gambetta, 76000 Rouen, France; 19Paul-Brousse Hospital, Public Hospsitals of Paris, 12 Avenue Paul Vaillant Couturier, FHU Hépatinov, 94800 Villejuif, France

**Keywords:** anti HCV therapy, DAAs, liver transplantation, decompensated cirrhosis, hepatocellular carcinoma, waiting list, recurrence after liver transplantation

## Abstract

Background: Direct-acting antiviral (DAA) agents for the treatment of hepatitis C virus (HCV) infection have been proven safe and effective in cirrhotic patients awaiting liver transplantation (LT). However, in the long term, data remain minimal regarding the clinical impact of viral eradication on patients listed for decompensated cirrhosis or hepatocellular carcinoma (HCC). We aimed to elucidate the clinical outcomes of patients regarding delisting and the evolution of HCC during the long-term follow-up. Methods: An observational, multicenter, retrospective analysis was carried out on prospectively collected data from HCV-positive patients treated with an interferon-free regimen while awaiting LT in 18 French hospitals. Results: A total of 179 patients were included in the study. The indication for LT was HCC in 104 (58.1%) patients and cirrhosis in 75 (41.9%) patients. The sustained virological response was 84.4% and the treatment was well tolerated. At five years, among 75 patients with cirrhosis treated for HCV, 19 (25.3%) were delisted following improvement after treatment. Predictive factors for delisting highlighted an absence of ascites, MELD score ≤ 15, and Child–Pugh score ≤ 7. No patients with refractory ascites were delisted. Among patients with HCC, 82 (78.9%) were transplanted. The drop-out rate was low (6.7%) and few recurrences of HCC after LT were observed. Conclusions: DAAs are safe and effective in patients awaiting LT for cirrhosis or HCC. A quarter of patients with cirrhosis can be delisted because of clinical improvement. Predictive factors for delisting, as a result of improvement, may assist prescribers, before initiating HCV infection therapy in the long-term perspective.

## 1. Introduction

Hepatitis C virus (HCV) infection has long been one of the main causes of end-stage liver disease and indications for liver transplantation (LT) worldwide [1]. Nonetheless, this indication is becoming less common thanks to the introduction of direct-acting antiviral (DAA) agents. However, today, some patients are still diagnosed with cirrhosis or hepatocellular carcinoma (HCC) [2]. During the interferon (IFN) therapy era, all HCV-positive recipients who underwent LT had detectable viremia and experienced HCV re-infection shortly after liver surgery. Additionally, this involved a high number of adverse events [3,4]. The advent of DAAs has revolutionized therapeutic strategies for HCV-positive patients and particularly those awaiting LT.

The first report evaluating treatment for HCV infection (sofosbuvir and ribavirin) in 61 patients with decompensated cirrhosis or hepatocellular carcinoma listed for LT encouraged hepatologists to treat HCV infection before LT. This strategy helped to prevent a recurrence of HCV positivity after LT and led to improvements in liver function that sometimes resulted in delisting [5]. Afterwards, Belli et al. subsequently reported results from a European study designed at determining the probability of delisting after DAA therapy. Here, the cumulative incidences of inactivated and delisted patients at 60 weeks of treatment were 33% and 19.2%, respectively [6]. Likewise, another study involving 238 patients enrolled with HCC or decompensated cirrhosis found a similar delisting rate after treatment for HCV infection [7]. HCV infection treatment has thus proved to be safe and effective in patients with cirrhosis awaiting LT [8,9]. However, some issues remain unresolved and require further investigation.

Firstly, an improvement in liver function allowing for delisting is not always observed after HCV infection treatment; additional data are required for the identification of patients who could benefit from this treatment before LT [10,11]. Secondly, there are several contradictory studies in the literature with respect to HCC recurrence after DAA treatment [12,13,14,15]; the indications for HCV infection treatment in these patients awaiting LT need to be specified.

The aims of our study were therefore primarily to assess the long-term efficacy and safety of DAA antiviral therapy in patients awaiting LT with decompensated cirrhosis or HCC. Subsequently, we aimed to determine the probability of patient delisting, as well as elucidate the clinical outcomes of delisted patients and the evolution of HCC during the long-term follow-up.

## 2. Patients and Methods

### 2.1. Patients and Study Design

We performed an observational, multicenter, retrospective analysis of data prospectively collected from 18 hospitals throughout France. All HCV-positive patients having received antiviral therapy with an IFN-free regimen while awaiting LT between November 2013 and June 2015 were consecutively enrolled in this study. Data on comorbidities (diabetes, dyslipidemia, alcohol consumption, arterial hypertension, BMI) were collected.

### 2.2. Data Collection

The antiviral regimen and treatment duration were decided at the physician’s discretion based on current guidelines and treatment availability. Data on any prior treatments for HCV infection were also collected. For patients who failed an initial course of antiviral therapy with DAAs and then received a second DAA regimen, only the second antiviral regimen was analyzed. Liver and renal function tests, as well as blood sample data were analyzed at: each visit during DAA therapy, end of treatment (week 24), post-treatment week 12, post-treatment year 1, post-treatment year 2 and post-treatment year 5. MELD and Child–Pugh scores were calculated at each visit using standard formulas based on measured parameters. HCV RNA levels were monitored at baseline, at end of treatment, and at post-treatment week 12.

An end of treatment response was defined as undetectable HCV RNA levels at end of treatment. A sustained virological response (SVR) was defined as undetectable HCV RNA at week 12 after treatment discontinuation (before or after LT). All patients were monitored for adverse events at each visit. Clinical and biological responses of patients to antiviral treatment were defined according to changes in Child–Pugh score from between treatment start and 12 weeks after the end of treatment as follows: complete response: Child–Pugh B/C to A; partial response: Child–Pugh C to B; stable response: Child–Pugh A to A; no response: Child–Pugh B or C, or a deteriorating condition. In patients with decompensated cirrhosis, any decision to delist a patient was at the physician’s discretion. For patients with HCC, the number of nodules, the diameter of the largest nodule, and alpha-fetoprotein (AFP) levels were recorded before treatment, at the end of treatment, and at the end of follow-up (AFP score). Treatments for HCC were also recorded. The onset of HCC during follow-up was also reported for patients on the list with decompensated cirrhosis.

### 2.3. Liver Graft Allocation Procedure

The French liver transplantation allocation system is based on the MELD score. There are some exceptions for cases of hepatic complications despite a low MELD score. In a context of HCC, the criteria for LT are based on the AFP score, which must be ≤2, with the HCC at tumor-node-metastasis (TNM) classification stage 2 [16]. The average waiting time on the LT list for patients with HCC is 6–12 months [16]. In this study, patients were listed for transplantation for the indication of decompensated cirrhosis in case of high MELD or expert component for refractory ascites or hepatic encephalopathy for example. In the case of HCC, patients were listed for this indication.

### 2.4. Statistical Analysis

Descriptive statistics are presented as means (±SD) and medians (interquartile ranges) for quantitative variables and counts (percentages) for qualitative variables. The Wilcoxon rank sum test was applied to compare the distribution of continuous variables and Chi-squared test (or Fisher’s exact test when appropriate) was used to test the association of categorical variables. We performed a ROC curve to determine the optimal cut-off values of MELD and CHILD scores for delisting (maximum of Youden’s index). A multivariate Logistic regression was established to determine predictive factors for delisting (and also for LT) based on selected clinical and biological characteristics (univariate analysis; level of significance: *p*-value < 0.20). In addition, three procedures for selecting variables (forward, backward, and stepwise) were used to obtain the most appropriate logistic-regression model. All procedures led to the same final model. The Hosmer–Lemeshow test was used to assess the goodness of fit of the logistic model. In addition, Kaplan–Meier method was used to generate overall survival and LT-free survival estimates, and the log-rank test was applied to compare groups. A p-value <0.05 was considered to be statistically significant and all statistical tests were two-sided. All statistical analyses were performed using SAS software V.9.4 (SAS Institute, Cary, NC, USA).

## 3. Results

### 3.1. Patient Characteristics at Baseline

The flow chart presented in Figure 1 indicates that a total of 179 patients were included for study. The indication for LT was HCC in 104 (58.1%) patients and decompensated cirrhosis in 75 (41.9%) patients. The baseline characteristics of these patients at the start of treatment are described in Table 1. The majority of patients were male (145; 81.0%) with a median age of 54 years [IQR 51; 59 years]. Twelve (6.7%) patients had an HIV co-infection, while chronic alcohol consumption was associated with HCV infection in 39 (22.2%) patients. Some patients also presented with metabolic comorbidities: arterial hypertension (*n* = 44; 25.1%), diabetes (*n* = 48; 27.4%), and dyslipidemia (*n* = 8; 4.6%). In the overall study population, 59 (33.0%) patients presented with moderate or refractory ascites and 31 (17.3%) patients presented with encephalopathy. At baseline, the median MELD and Child–Pugh scores were 11 [IQR 8; 15] and 7 [IQR 5; 9], respectively.

### 3.2. Efficacy and Safety of Antiviral Therapy

Genotype 1 was the most common HCV infection. Sofosbuvir/daclatasvir was the most widely administered antiviral regimen (*n* = 111; 62.0%) (Appendix A). The overall SVR was 84.4%, with 151 patients being cured (69 patients with decompensated cirrhosis and 82 with HCC). Treatment was well tolerated, with serious adverse events (SAEs) occurring in 34 (19.0%) patients, including six deaths (Appendix A).

At the time of analysis, 121 (67.6%) patients had undergone LT: 82 (78.9%) from the HCC group and 39 (52.0%) from the decompensated cirrhosis group. Among patients who did not undergo LT, 19 were delisted for an improvement and 29 for a deterioration in their condition. Ten patients were still inactive on the transplant list at the end of follow-up. The median follow-up period was 84.5 months (95% CI; 83.6–85.4). Forty-four (24.6%) patients died during follow-up, but there was no significant difference in overall survival between patients enrolled for HCC or decompensated cirrhosis (77.3% vs. 74.0% for cirrhosis and HCC, respectively; *p* = 0.63) (Figure 2A). Transplant-free survival was significantly longer among patients with decompensated cirrhosis compared to those with HCC (42.4 vs. 12.1 months, respectively; *p* = 0.0001) (Figure 2B).

### 3.3. Patients Subgroup with Decompensated Cirrhosis

Seventy-five patients were enrolled for study in a context of decompensated cirrhosis. Their clinical and biological characteristics at baseline, during treatment, and after treatment are presented in Table 2. At baseline, the median MELD and Child–Pugh scores were 14 [IQR 11; 18] and 8 [IQR 7; 10], respectively. Thirty-nine (52.0%) patients were transplanted following an interval between the end of treatment and LT of 8.9 (± 13.4) months. In three of these patients, transplant surgery was performed before the end of treatment and treatment was pursued after LT. Among the patients who were not transplanted, 31 were delisted: 19 as a result of improvement and 12 due to deterioration in their condition or contraindication. The rate of delisting for improvement was 25.3%. Three patients delisted for improvement died during the follow-up period. Twelve additional patients were delisted for the following indications: HCC not meeting criteria for LT (*n* = 4), bladder cancer (*n* = 1), lost to follow-up or refusal by the patient (*n* = 2), sepsis (*n* = 1), extensive portal thrombosis (*n* = 1), road accident (*n* = 1), or deterioration in condition (*n* = 2). Nine of these 12 delisted patients subsequently died (Figure 1). Treatment of HCV infection resulted in a complete clinical and biological response in 13 (24.5%) patients, defined by an improvement to Child–Pugh score A without hepatic encephalopathy or ascites. Fifteen (28.3%) patients had stable or partially improved clinical or biological liver disease after treatment of their HCV with DAAs. Twenty-five (47.2%) patients did not show a response, or their clinical and biological condition deteriorated. The optimal cut-off values for MELD and Child–Pugh scores were determined by ROC curve analysis in order to predict delisting due to improvement. Accordingly, the Youden’s index indicated that the optimal cut-off point for the MELD score to delist a patient for improvement was 15.0 (95% CI; 12.1-23.0) (Figure 3A, Appendix A). For the Child–Pugh score, the optimal cut-off was calculated at 7 (95% CI; 5–9) (Figure 3B, Appendix A).

### 3.4. Predictive Factors for Delisting or LT

Concerning predictive factors for delisting as a result of improvement, univariate analysis calculated the MELD score, Child–Pugh score, absence ascites, bilirubin, and prothrombin time (PT). Under multivariate analysis, only the MELD score (OR: 0.820 [0.710–0.949]) was found to be an independent predictive factor of delisting for improvement. Furthermore, no patients with refractory ascites were delisted (Table 3a).

Predictive factors for LT were also analyzed. Following univariate and multivariate analysis, we identified the following liver transplantation predictive factors: HCC (OR: 7.32 [3.077–17.431]); bilirubin (OR: 1.025 [1.008–1.042]), and refractory ascites (OR: 0.275 [0.076–0.998]) (Table 3b).

### 3.5. Progression and Recurrence of HCC

One hundred and four patients were included for study in a context of HCC (Figure 1). Their clinical and biological characteristics at baseline, during, and after treatment are presented in Table 1. At baseline, the median MELD and Child–Pugh scores were 9 (IQR 7; 12) and 6 (IQR 5; 7), respectively. At the start of HCV infection treatment, 75 (93.8%) patients met the LT criteria for HCC (AFP score ≤ 2) and five patients had an AFP score > 2. For 24 patients out of the 104 (23.1%), the data available did not allow for calculation of the AFP score. Eighty-two (78.9%) patients were transplanted following a mean interval of 7.7 (± 11.6) months between the end of treatment and LT. Fifteen patients died post-LT and we observed five recurrences of HCC. Among the 22 patients who were not transplanted, 17 were delisted: seven for HCC progression (drop-out rate: 6.7%). The other indications for delisting were: improvement (*n* = 1), LT refusal (*n* = 2), extra-hepatic cancer (*n* = 1), alcohol relapse (*n* = 3), and others (*n* = 3). Twelve of the 22 not transplanted patients subsequently died. Among the 82 transplanted patients, 67 were still alive at the last follow-up. The treatments most frequently administered were trans-arterial chemoembolization (TACE) and radiofrequency ablation.

## 4. Discussion

Treatment for chronic HCV infection improves liver function in patients with decompensated cirrhosis. This improvement is not consistent and is sometimes insufficient for avoiding LT in these patients. Predictive factors for delisting or LT are therefore necessary to decide whether or not to initiate HCV infection treatment. The problem is slightly different for patients awaiting LT for HCC, but likewise, the indication and timing for HCV infection treatment still needs to be clarified for these patients [17,18]. Our study involved a large French multicenter cohort including 179 patients with HCV infection and awaiting LT. These patients presented both with and without HCC and had a long post-treatment follow-up.

The results concerning overall survival at five years were outstanding (75.4%). The tolerance of DAA therapy was satisfactory (SAEs: 19%) and comparable to the recently reported results in the real-world experience of the HCV-TARGET cohort [9]. Overall SVR was also high (84.4%) as previously described in other study cohorts [6,7,9,19]. However, we noted that SVR results were higher in our decompensated cirrhosis group (92%) than in our HCC group (78.9%). The issue of DAA efficacy in patients with HCC has already been discussed in the study by Beste et al. [17,18]. We could explain these higher results in our cohort by the observation that approximately half of our patients were treated with ribavirin and more than 74% of our overall study cohort had been previously treated. Note that the AADs currently used in these patients (Sofosbuvir/Velpatasvir) are slightly different and maybe more efficient than those in our cohort.

In our study, the delisting rate as a result of improvement in patients with decompensated cirrhosis was 25.3%. This is comparable to previously reported findings ranging from 19.2%–30.9% [6,7,20]. After over five years of follow-up, 3 patients of the 19 delisted patients died (2 for HCC, not eligible for LT because too old and macrovascular invasion). Minimal data are available regarding the long-term follow-up of patients with decompensated cirrhosis treated for HCV infection. Nonetheless, in a recent study comprising over four years of follow-up, 29% of patients experienced a clinically meaningful improvement in their MELD scores (by 3 or more points) and 25% achieved a MELD score < 10 [9]. These results are consistent with those in our study and thus confirm that treating HCV infection in patients with decompensated cirrhosis enables transplantation avoidance in one out of four (25.3%) patients. Among the predictive factors for delisting, we wish to highlight the absence of ascites, MELD score ≤ 15, and Child–Pugh score ≤ 7. No patients with refractory ascites were delisted in our study. However, it is worth noting that our study was performed before an improvement in survival was demonstrated after transjugular intrahepatic portosystemic shunt (TIPS) for treatment of refractory ascites [21]. In previous studies regarding HCV infection treatment of patients with decompensated cirrhosis, a MELD score of 16 was found to be predictive of delisting for improvement [6,9]. On the other hand, a cost-effectiveness study found a MELD threshold of 20 was necessary to decide on HCV infection treatment before or after LT [22]. Schaubel et al. evaluated the five-year survival benefit of LT in patients with chronic liver disease. In this study, the authors found a survival benefit at a MELD score of >10 [23]. Although the MELD 15 threshold for delisting found in our study seems low, it is consistent with the results assessing long-term survival in chronic liver disease patients.

Among the different study cohorts found in the literature, most patients have been enrolled on LT lists for decompensated cirrhosis, but data are scarce with respect to patients on LT lists for HCC, and particularly with a long follow-up [6,7,9]. In our study, 104 patients were listed for HCC and 78.9% were transplanted. In France, the transplant criteria for HCC use the AFP score (≤2) [16]. Hence, the low recurrence rate of HCC after liver transplantation (only five patients) confirmed the value of this score. The drop-out rate was also low in our study. There are indeed several arguments in the literature in favor of post-transplant HCV infection treatment of patients with HCC: better cost effectiveness [22], lower SVR rates in patients with HCC [18], and a risk of pre-transplant HCV infection treatment lengthening the waiting time on the LT list due to improved liver function [24]. These arguments must additionally be balanced against the possibility of having access to waiting list times for patients with HCC.

There are limitations to our study inherent to its retrospective nature and study design, and notably to the lack of data available for determining changes to Child–Pugh and MELD scores after HCV infection treatment. Another issue arises with respect to the classification of patients as “inactive on the list” or “delisted” given this is an arbitrary decision that may vary from one LT center to another.

In summary, this large French patient cohort under HCV infection treatment while awaiting LT for decompensated cirrhosis or HCC with a long-term follow-up confirms the high efficacy and safety of antiviral therapy with DAAs. Overall, HCV infection treatment allows for LT avoidance in one out of four (25.3%) patients with decompensated cirrhosis, particularly for cases with absence of ascites, MELD score ≤ 15, and Child–Pugh score ≤ 7. The risk is insufficient improvement in liver function that can result in patients falling in a situation of “MELD purgatory” [24]. The results of our current study also tend to be in favor of treating HCV infection before LT in patients with HCC given that we demonstrate a low drop-out rate and few recurrences of HCC post-LT. These findings will assist prescribers in their decisions regarding HCV infection treatment in patients awaiting LT.

## Figures and Tables

**Figure 1 viruses-15-00137-f001:**
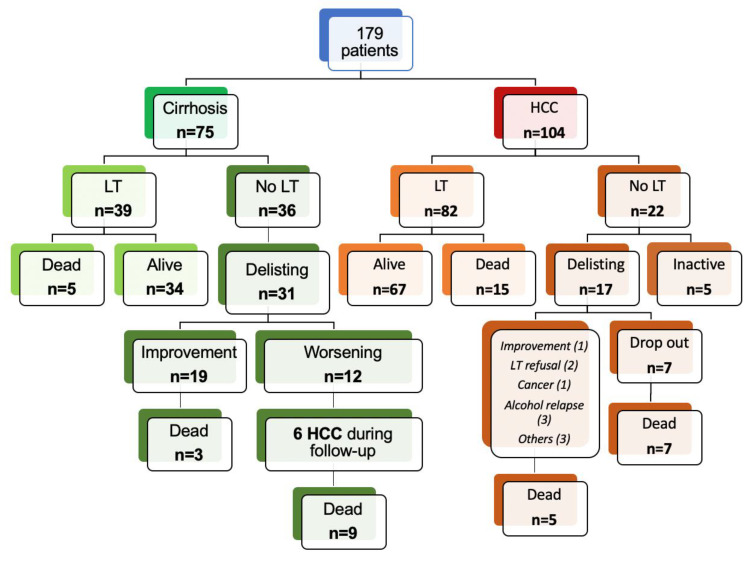
Description of the cohort. HCC: hepatocellular carcinoma; LT: liver transplantation.

**Figure 2 viruses-15-00137-f002:**
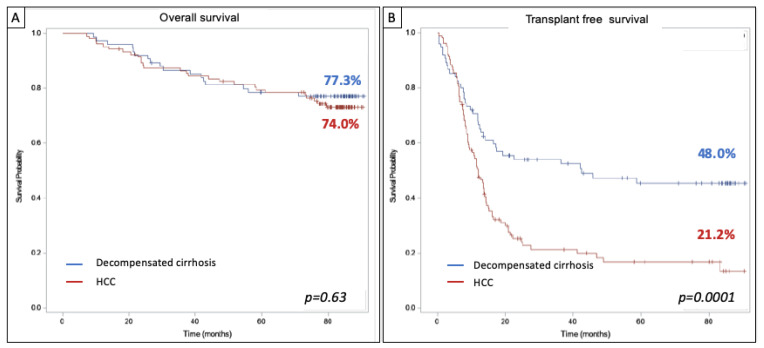
(**A**) Overall survival and (**B**) Transplant-free survival among patients awaiting LT listed for HCC (*n* = 104) or decompensated cirrhosis (*n* = 75) since HCV treatment.

**Figure 3 viruses-15-00137-f003:**
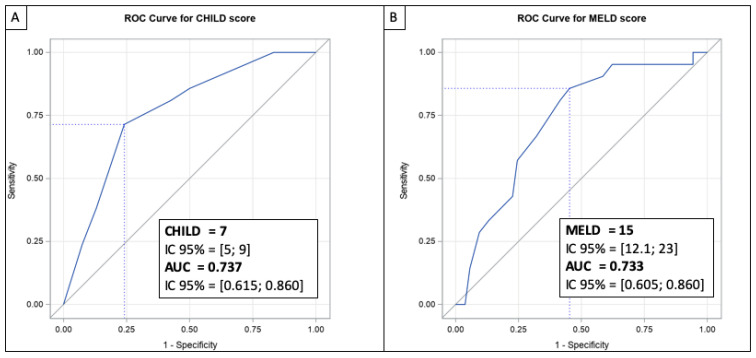
ROC curve for Child–Pugh score (**A**) and ROC curve for MELD score (**B**) to predict delisting after HCV treatment for an improvement in patients awaiting LT for decompensated cirrhosis.

**Table 1 viruses-15-00137-t001:** Baseline characteristics of the 179 patients awaiting LT and treated for HCV.

	Overall *n* = 179 (%)	Decompensated Cirrhosis *n* = 75 (%)	HCC *n* = 104 (%)
Male	145 (81)	57 (76.0)	88 (84.6)
Age (years) Median (IQR)	54 (51; 59)	53 (49; 57)	54.5 (52; 60)
BMI (kg/m^2^) Median (IQR)	25.3 (22.6; 28)	25.2 (22.8; 28)	25.3 (22.2; 28)
Diabetes Mellitus	48 (27.4)	25 (34.3)	23 (22.6)
Arterial hypertension	44 (25.1)	13 (17.8)	31 (30.4)
Alcohol consumption	39 (22.2)	14 (18.9)	25 (24.5)
Dyslipidemia	8 (4.6)	2 (2.7)	6 (5.9)
Ascites None Controlled Refractory	120 (67) 35 (19.6) 24 (13.4)	33 (44) 25 (33.3) 17 (22.7)	87 (83.7) 10 (9.6) 7 (6.7)
Hepatic encephalopathy None Grade I-II	148 (82.7) 31 (17.3)	50 (66.7) 25 (33.3)	98 (94.2) 6 (5.8)
Total bilirubin(μmol/L) Median (IQR)	29.8 (16; 48)	41 (25.3; 62)	19 (12; 35)
INR Median (IQR)	1.2 (1.1; 1.5)	1.5 (1.3; 1.8)	1.2 (1.1; 1.2)
Albumin (g/L) Median (IQR)	33.1 (29; 37.6)	30 (26; 35.9)	35.3 (31; 38.7)
MELD baseline Median (IQR)	11 (8; 15)	14 (11; 18)	9 (7; 12)
Child–Pugh class at baseline A B C	83 (48) 51(29.5) 39 (22.5)	15 (20) 30 (40) 30 (40)	68 (69.4) 21 (21.4) 9 (9.2)
SVR rate	151 (84.4)	69 (92)	82 (78.9)
Liver transplant recipients	121 (67.6)	39 (52)	82 (78.9)

LT: liver transplantation; HCV: hepatitis C virus; HCC: hepatocellular carcinoma; BMI: body mass index; INR: international normalized ratio; MELD: Model for End-Stage Liver Disease; SVR: sustained virological response.

**Table 2 viruses-15-00137-t002:** Hepatic function outcomes during and after HCV treatment in patients awaiting LT for decompensated cirrhosis.

Variable	Baseline	Week 24	Week 12 after End of Treatment	One Year Post-Treatment	Two Years Post-Treatment
MELD score Median (IQR)	14 (11; 18)	13 (9.5; 16)	12 (9; 15)	11 (8; 13)	9 (8; 12)
Child Pugh class, *n* (%)					
A	15 (20)	20 (37.7)	32 (49.2)	31 (70.5)	29 (78.4)
B	30 (40)	21 (39.6)	22 (33.9)	8 (18.2)	8 (21.6)
C	30 (40)	12 (22.6)	11 (16.9)	5 (11.4)	0 (0)

HCV: hepatitis C virus; LT: liver transplantation; MELD: Model for End-Stage Liver Disease.

**Table 3 viruses-15-00137-t003:** Predictive factors for delisting for an improvement in decompensated cirrhosis (a) and predictive factors for liver transplantation (b).

(a)		
Univariate Analysis		Multivariate Analysis
Variable	*p* value (test)	
MELD score	0.0026	OR (95% CI): 0.820 (0.710–0.949)
Child–Pugh score	0.0019	
Child–Pugh class	0.0215	
Bilirubin	0.0038	
Ascite	0.0213	
Albumin	0.0612	
PT	0.0083	
INR	0.0163	
HTA	0.1623	
Encephalopathy	0.1888	
**(b)**		
**Univariate analysis**		**Multivariate analysis**
Variable	*p* value (test)	
HCC	0.0002	OR (95% CI): 7.324 (3.077–17.431)
Bilirubin	0.1470	OR (95% CI): 1.025 (1.008–1.042)
ascites	0.0298	OR (95% CI): 0.275 (0.076–0.998)
Viral Load	0.0422	
Child–Pugh class	0.1312	

PT: prothrombin time; MELD: Model for End-Stage Liver Disease; HCC: hepatocellular carcinoma; OR: Odds ratio; test: χ2 or Wilcoxon rank sum.

## Data Availability

Available on request.

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
