# Peer review of "Patients Treated for HCV Infection and Listed for Liver Transplantation in a French Multicenter Study: What Happens at Five Years?"

_viruses, 2022, doi:10.3390/v15010137_

Round 1

Reviewer 1 Report (Previous Reviewer 2)

Comments adequately addressed.

Reviewer 2 Report (New Reviewer)

• The primary output/endpoint variable(s)/measurement(s) of the study should be defined.  • How was the sample size determined? This information should be explained in the Materials and Methods section. • How were extreme/outlier values in the data determined and resolved?  • What approaches were used to test the validity of the models? • Which metrics were used in the performance evaluation of the estimates of models/algorithms?  • How were the predictive models selected in this study? • Which method(s) was/were used to optimize the hyperparameters of models/algorithms? • How was the most suitable cut-off point determined using the receiver operator characteristic (ROC) curve analysis? • The number of current references on the subject of the study should be increased.

This manuscript is a resubmission of an earlier submission. The following is a list of the peer review reports and author responses from that submission.

Round 1

Reviewer 1 Report

This paper provides further insight on the effectiveness of DAAs in patients waiting LT for cirrhosis or HCC. Findings well presented and discussed. Correct acknowledgement of some limits of the study. I don't have any query.

Author Response

Answers point-by point to the comments of the reviewers

Patients treated for HCV infection and listed for liver trans-plantation in a French multicenter study: what happens at five years?

  1. Meunier, M Belkacemi, GP. Pageaux, S. Radenne, A. Vallet-Pichard, P. Houssel-Debry, C. Duvoux, D. Botta-Fridlund, V. de Ledinghen, F. Conti, R. Anty, V. Di Martino, M. Debette-Gratien, V. Leroy, T. Gerster, P. Lebray, L. Alric, A. Abergel, J. Dumortier, C. Besch, H. Montialoux, D. Samuel, JC. Duclos-Vallée and A. Coilly

Many thanks for the time and effort from yourselves and the reviewers and the valuable feedback on this manuscript. We have now addressed these comments in full, and point-by-point responses are appended below. In addition, major changes are highlighted in red in the revised version of the manuscript.

We hope that you agree that the manuscript is significantly improved as a result of these changes, and that it is now suitable for publication in Viruses.

Reviewer 1

This paper provides further insight on the effectiveness of DAAs in patients waiting LT for cirrhosis or HCC. Findings well presented and discussed. Correct acknowledgement of some limits of the study. I don't have any query.

Reviewer 2

The authors provide an interesting study of outcomes of liver transplant candidates with cirrhosis vs. HCC who were treated for HCV infection in the early era of DAAs. The patients are drawn from a prospective cohort study; no control group not receiving DAAs is presented, so limited comparisons are possible, significantly limiting the interpretability of results.

  • There are grammatical errors throughout that compromise understanding and should be corrected (e.g. in abstract “75 patients with cirrhosis cures for HCV”, or in section 3.4 “with respect to predictive factors for delisting as a result of improvement, univariate analysis calculated MELD score”)

Thank you for pointing out these errors to help us improve the quality of the article. English proofreading was done by a non-medical professional. A new proofreading was done. The following corrections have been made to the manuscript : “75 patients with cirrhosis cures for HCV” modified for “75 patients with cirrhosis treated for HCV” and “with respect to predictive factors for delisting as a result of improvement, univariate analysis calculated MELD score” modified for ”Concerning predictive factors for delisting as a result of improvement, univariate analysis calculated MELD score”.

  • Since there is no control group in this study, providing comparisons for % typically delisted for clinical improvement and % typically delisted for deterioration in France would greatly improve interpretability of results

Thank you for this very relevant comment. We extracted the following data from the French organ donation registry during the same inclusion period as our study. https://www.agence-biomedecine.fr/annexes/bilan2015/donnees/organes/05-foie/synthese.htm

Years

Patients on waiting list

Nb liver transplant

Nb of delisting

Delisting for deterioration

2013

1820

1241

212

116

2014

1786

1280

262

138

2015

1756

1355

251

122

During the same period, there was a decrease in the number of patients on the list and an increase in the number of LT. During this period the number of patients discharged for improvement has increased, possibly related to the effect of HCV in treated patients.

  • Given that these patients received DAAs in the early days of this medical therapy (2013-2015), please discuss how treatment protocols have changed, how this might affect results (including rate of sustained virologic response), and generalizability to patients currently on the waitlist.

You are right, treatments have evolved since the inclusion period of our study. In our study, the patients were all treated with sofosbuvir, more than half of them with daclatasvir. Half of the patients also had ribavirin (cf table S1). The SVR rate in our study was 84.4% (151/179 patients) but 92% in decompensated cirrhotic group.

Currently, the reference treatment for HCV in decompensated cirrhotic patients is Sofosbuvir/Velpatasvir.

In studies, the SVR with this treatment is 94% (95% CI, 87 to 98) among those who received 12 weeks of sofosbuvir-velpatasvir plus ribavirin, and 86% (95% CI, 77 to 92) among those who received 24 weeks of sofosbuvir-velpatasvir, which is higher than with the previous treatments (PMID: 26569658).

However, to the best of our knowledge, there are no studies comparing the effectiveness of older DAA and newer DAA regimens. In our cohort, SVR is lower for HCC patients.

You are right, this difference in effectiveness should be considered when generalising our results to patients currently on the list.

This remark has been added in the discussion of the article “Note that the AADs currently used in these patients (Sofosbuvir/Velpatasvir) are slightly different and maybe more efficient, from those in our cohort”.

  • In Section 3.1, please clarify whether the bracketed numbers provided after median values are the full range (as in Table 1) or interquartile range (as is more commonly presented).

The manuscript has been corrected with median and interquartile range.

  • For continuous variables in Table 1, I would recommend choosing either mean (SD) or median (IQR) to represent the data, based on whether they are normally distributed or not.

The manuscript has been corrected by keeping the medians with interquartile range.

  • The authors use multivariable logistic regression and limit their time-to-event analysis to Kaplan-Meier curves and log-rank tests. I would strongly urge them to consider additional time-to-event analyses (i.e. multivariable Cox regression and Fine-Gray competing risk analysis) given that this is not a closed cohort. 

Thank you for this important remark, unfortunately we do not have the data concerning the delisting dates. This is a real life study and delisting policies vary from centre to centre.

  • In Section 3.2, the percentage of patients that have undergone LT at the time of analysis is presented, seemingly ignoring the differences in time on the waitlist that patients have accrued. It does not appear that all patients' follow-up time was truncated at 5 years, given that the discussion mentions patients "at least 5 years post-LT". I would recommend looking at percentage transplanted by a particular timepoint (e.g. 1 year, 3 years, 5 years) if the authors wish to provide such a number, but overall would recommend a time-to-event method.

The following are data on the transplant rate as a function of time.

Product-Limit Estimates

Timelist

Number
Transplant

Transplant

Rate

IC95%

1 year

75

0.42

(0.35 ; 0.50)

3 years

112

0.64

(0.57 ; 0.72)

5 years

120

0.71

(0.63 ; 0.78)

  • In Figures 2A and 2B, please clarify on the x-axis that time is time since listing, not time since HCV treatment

The x-axis is the time since HCV treatment. We added this information to the figure legend.

We chose this date because it seemed more robust to show the effect of HCV treatment.

Indeed, the sequence of HCV treatment and listing was not homogeneous for all patients.

Depending on the policies/habits of the transplant centres some patients have been treated for HCV before or after being listed for transplantation (see next point).

  • Please include more information about the timing of treatment for HCV; for most patients, was this before or during listing? For patients delisted for improvement, how soon after HCV treatment did this typically occur?

For most patients (128 patients) HCV treatment was after listing, on average 321 days later (min : 0 day and max : 2743 days). For 51 patients, HCV treatment was prior to listing (mean: 112 days, min : 0 day, max: 425 days) . However, these patients were in the process of liver transplantation and were being screened for LT at the time of HCV treatment. Concerning the delay between HCV treatment and delisting we unfortunately do not have this information because we do not have the dates of the patients' delisting. Delisting policies differed greatly between centres, with some choosing to keep the patient inactive on the list for a period of time before delisting.

  • Please add additional information about the Kaplan-Meier analysis for transplant-free survival. In this case, is the event of interest death, with censoring for transplant or delisting? How does this analysis treat patients who were delisted for decompensation and subsequently died?
  • In Section 3.3, please list the number delisted for deterioration and the number delisted for contraindication separately, for clarity. Please also clarify if the patients who were delisted and subsequently died were all from the group listed for deterioration and contraindication (the new denominator of 10 suggests it might be the group delisted for contraindication); if so, it is not surprising that they died, but this would be surprising if they were delisted for improvement.

Thank you for pointing out this inconsistency. After checking our data, it is indeed 12patients delisted for deterioration and 9 deceased as described in figure 1. Concerning the 3 non-deceased patients out of 12, 2 are lost to follow-up and 1 still alive with bladder cancer.

Regarding the patients who died after delisting for improvement, the causes of death were: 2 palliative CHC, 1 hypoglycaemic coma. The manuscript has been modified.

  • In section 3.3, please clarify the sentence that states there was a “stable or partial response to treatment of HCV” to indicate that you are speaking about their overall liver disease and its reversibility, not the response of their HCV to treatment. Different terminology, such as “15 (28.3%) patients had stable or partially improved clinical or biological liver disease after treatment of their HCV with DAAs” would improve this sentence.

Corrections have been included in the revised manuscript.

  • In Table 2, how was week 24 chosen? Is this week 24 after listing? Selection of this time point is not explained/justified in the Methods section.

The 24-week time point was chosen because it corresponded for most patients to the end of

HCV treatment. The data at W12 seemed too early to assess the effect of the HCV treatment.

Precisions have been added to the methods.

  • Given that at least half of the patients in your study had a MELD score <=15 at ever timepoint (including listing), the threshold you establish for optimal delisting, how should their MELD score be used to inform delisting?

Indeed, the MELD threshold predictive of delisting is very low in our study but it is a long-term prediction. In liver transplantation studies that have looked at the benefit of TH according to MELD, the longer the interval, the lower the MELD as shown in the figure below.

PMID: 19341419

  • For the analyses regarding predictive factors for delisting, were both HCC and cirrhosis patients included? Since only one patient with HCC was delisted for improvement and there are numerous baseline differences between the HCC and cirrhosis populations in this study, these results might not be truly generalizable to the HCC population. Also, was only the categorical MELD dichotomized at the median (as described in section 2.4) used in this analysis? Did the authors explore models with more categories of MELD, since a MELD of 15 is where where survival benefit from transplant first occurs in some studies?

Only patients enrolled with decompensated cirrhosis were considered for delisting risk factors. Indeed we considered that the delisting factors were not applicable to patients with HCC.

Indeed we have explored the model with different categories of MELD without any significant results.

  • Regarding section 3.5, why were patients who did not meet LT criteria for HCC included? Did these patients receive transplants; if so, were they among those who experienced recurrence and/or died?

Five patients were outside the AFP criteria initially (at the start of HCV treatment) but subsequently downstaged:

- 3 patients LT and alive, no recurrence

- 1 patient delisted for worsening / progression of HCC

- 1 patient died post LT for infection (LT outside criteria and death 1 year after LT for sepsis, no notion of HCC recurrence)

  • Were any patients who were delisted for improvement subsequently relisted for transplant consideration? Was this treatment option considered for the patients with cirrhosis who were delisted and subsequently developed HCC? (see line 272)

To the best of our knowledge, no delisted patients were subsequently relisted.

Of the 3 patients delisted for improvement and died, 2 had HCC. For these 2 patients LT was not proposed for the following reasons:

- patient older than 70 years at the time of diagnosis

- HCC with macrovascular invasion

These clarifications were included in the manuscript.

  • Line 298 – you state that you found no recurrence of HCC after HCV treatment, while in the previous sentence you state that 5 patients had an HCC recurrence rate after liver transplant and in the abstract you state that “few recurrences of HCC after LT was observed”. Please clarify.

Thank you for pointing out this discrepancy.

The sentence «  We did not find any recurrence of HCC after HCV infection treatment, which is contrary to outcomes suggested in some previously reported studies “ has been removed.

We confirm that there were 5 recurrences of HCC post LT.

Reviewer 2 Report

The authors provide an interesting study of outcomes of liver transplant candidates with cirrhosis vs. HCC who were treated for HCV infection in the early era of DAAs. The patients are drawn from a prospective cohort study; no control group not receiving DAAs is presented, so limited comparisons are possible, significantly limiting the interpretability of results.

- There are grammatical errors throughout that compromise understanding and should be corrected (e.g. in abstract “75 patients with cirrhosis cures for HCV”, or in section 3.4 “with respect to predictive factors for delisting as a result of improvement, univariate analysis calculated MELD score”)

- Since there is no control group in this study, providing comparisons for % typically delisted for clinical improvement and % typically delisted for deterioration in France would greatly improve interpretability of results

- Given that these patients received DAAs in the early days of this medical therapy (2013-2015), please discuss how treatment protocols have changed, how this might affect results (including rate of sustained virologic response), and generalizability to patients currently on the waitlist.

- In Section 3.1, please clarify whether the bracketed numbers provided after median values are the full range (as in Table 1) or interquartile range (as is more commonly presented).

- For continuous variables in Table 1, I would recommend choosing either mean (SD) or median (IQR) to represent the data, based on whether they are normally distributed or not.

- The authors use multivariable logistic regression and limit their time-to-event analysis to Kaplan-Meier curves and log-rank tests. I would strongly urge them to consider additional time-to-event analyses (i.e. multivariable Cox regression and Fine-Gray competing risk analysis) given that this is not a closed cohort. 

- In Section 3.2, the percentage of patients that have undergone LT at the time of analysis is presented, seemingly ignoring the differences in time on the waitlist that patients have accrued. It does not appear that all patients' follow-up time was truncated at 5 years, given that the discussion mentions patients "at least 5 years post-LT". I would recommend looking at percentage transplanted by a particular timepoint (e.g. 1 year, 3 years, 5 years) if the authors wish to provide such a number, but overall would recommend a time-to-event method.

- In Figures 2A and 2B, please clarify on the x-axis that time is time since listing, not time since HCV treatment

- Please include more information about the timing of treatment for HCV; for most patients, was this before or during listing? For patients delisted for improvement, how soon after HCV treatment did this typically occur?

- Please add additional information about the Kaplan-Meier analysis for transplant-free survival. In this case, is the event of interest death, with censoring for transplant or delisting? How does this analysis treat patients who were delisted for decompensation and subsequently died?

- In Section 3.3, please list the number delisted for deterioration and the number delisted for contraindication separately, for clarity. Please also clarify if the patients who were delisted and subsequently died were all from the group listed for deterioration and contraindication (the new denominator of 10 suggests it might be the group delisted for contraindication); if so, it is not surprising that they died, but this would be surprising if they were delisted for improvement.

- In section 3.3, please clarify the sentence that states there was a “stable or partial response to treatment of HCV” to indicate that you are speaking about their overall liver disease and its reversibility, not the response of their HCV to treatment. Different terminology, such as “15 (28.3%) patients had stable or partially improved clinical or biological liver disease after treatment of their HCV with DAAs” would improve this sentence.

- In Table 2, how was week 24 chosen? Is this week 24 after listing? Selection of this time point is not explained/justified in the Methods section.

- Given that at least half of the patients in your study had a MELD score <=15 at ever timepoint (including listing), the threshold you establish for optimal delisting, how should their MELD score be used to inform delisting?

- For the analyses regarding predictive factors for delisting, were both HCC and cirrhosis patients included? Since only one patient with HCC was delisted for improvement and there are numerous baseline differences between the HCC and cirrhosis populations in this study, these results might not be truly generalizable to the HCC population. Also, was only the categorical MELD dichotomized at the median (as described in section 2.4) used in this analysis? Did the authors explore models with more categories of MELD, since a MELD of 15 is where where survival benefit from transplant first occurs in some studies?

- Regarding section 3.5, why were patients who did not meet LT criteria for HCC included? Did these patients receive transplants; if so, were they among those who experienced recurrence and/or died?

- Were any patients who were delisted for improvement subsequently relisted for transplant consideration? Was this treatment option considered for the patients with cirrhosis who were delisted and subsequently developed HCC? (see line 272)

- Line 298 – you state that you found no recurrence of HCC after HCV treatment, while in the previous sentence you state that 5 patients had an HCC recurrence rate after liver transplant and in the abstract you state that “few recurrences of HCC after LT was observed”. Please clarify.

Author Response

(The authors gave the same response as above.)
